# Soft Multi-Directional Force Sensor for Underwater Robotic Application

**DOI:** 10.3390/s22103850

**Published:** 2022-05-19

**Authors:** Rafsan Al Shafatul Islam Subad, Md Mahmud Hasan Saikot, Kihan Park

**Affiliations:** 1Department of Mechanical Engineering, University of Massachusetts Dartmouth, 285 Old Westport Road, Dartmouth, MA 02747, USA; rsubad@umassd.edu; 2Department of Mechanical Engineering, Bangladesh University of Engineering and Technology, Dhaka 1000, Bangladesh; saikath65@gmail.com

**Keywords:** soft tactile sensor, flex sensor, multi-directional force sensing, underwater soft robotics

## Abstract

Tactile information is crucial for recognizing physical interactions, manipulation of an object, and motion planning for a robotic gripper; however, concurrent tactile technologies have certain limitations over directional force sensing. In particular, they are expensive, difficult to fabricate, and mostly unsuitable for underwater use. Here, we present a facile and cost-effective synthesis technique of a flexible multi-directional force sensing system, which is also favorable to be utilized in underwater environments. We made use of four flex sensors within a silicone-made hemispherical shell structure. Each sensor was placed 90${}^{{}^{\circ}}$ apart and aligned with the curve of the hemispherical shape. If the force is applied on the top of the hemisphere, all the flex sensors would bend uniformly and yield nearly identical readings. When force is applied from a different direction, a set of flex sensors would characterize distinctive output patterns to localize the point of contact as well as the direction and magnitude of the force. The deformation of the fabricated soft sensor due to applied force was simulated numerically and compared with the experimental results. The fabricated sensor was experimentally calibrated and tested for characterization including an underwater demonstration. This study would widen the scope of identification of multi-directional force sensing, especially for underwater soft robotic applications.

## 1. Introduction

Having the presence of a mechanoreceptor that enables sensing both the direction and the magnitude of the applied force on the fingertip instantaneously, human hands can manipulate objects and perform a variety of tasks deftly [1]. Robotic grippers analogous to human fingers nowadays are projected to handle differently shaped objects, adapting the grasp, and in addition, are able to execute their tasks in an unstructured environment [2,3,4]. Although soft robotic end effectors allow manipulators to overcome the gap between known and unexpected morphologies, operating these systems remotely with high accuracy and precision still remains a challenging problem to be addressed extensively [5,6]. Today’s typical underwater robots, for example, rely heavily on camera sensing capabilities in the visible area for detailed imaging and recording [7]; however, there exist numerous instances where an optical camera might not work properly due to the low visibility in the larger part of the underwater atmosphere. Acoustic sensors are one of the viable options for those scenarios but they are low-resolution ranging, and often provide noisy and distorted signals [8]. Tactile sensors, as a merged solution, can surely play pivotal roles in terms of sensing, controlling, and manipulating an object locally with high sensitivity and spatial resolution, especially in opaque underwater environments [9,10,11]. Moreover, targeted applications such as intact sample collections, explosive ordinance disposal (EOD) missions, repairing of unmanned underwater vehicles (UUVs), autonomous underwater vehicles (AUVs), etc., underwater require physical contact with objects [12,13,14]. In these circumstances, the integration of haptic feedback by gathering local tactile information most importantly, the identification of multi-directional force leads to an improvement in dimensional awareness and is kind of a fundamental requirement to achieve those goals [15,16]. Nowadays, commercial-grade force sensors are mostly normal force sensors. Fabrication of a soft tactile sensor capable of detecting multi-directional forces is still an active research area [17,18]. Thus, it is essential to develop a cost-effective and readily synthesizable multi-directional force sensing system that can be compared with human perception in terms of force recognition for advanced robotic manipulation.

At present, a broad range of solutions is available for tactile sensor technology depending on different types of hardware employed [19,20]; however, there are only a few studies for flexible sensors that can detect both normal and tangential forces [1]. Teshigawara et al. demonstrated a slip detection system for a multi-fingered robot using pressure-sensitive, conductive rubber [21]. The system not only distinguishes normal forces but also offers information about shear forces and slippage. Harada et al. presented an approach of utilizing resistivity to recognize touch, and slip/friction in their study [22]; however, their sensor was limited to a tactile force in one direction because of the difficulties fabricating devices on flexible substrates. Lee et al. demonstrated a flexible tactile sensor array based on the principle of capacitor with the capability of measuring both normal and shear force [23]. They used polydimethylsiloxane (PDMS) for the sensor structure and it can measure up to 131 kPa in three directions. Huang et al. and Zhang et al. also made use of a similar capacitive methodology to evaluate multi-directional forces in their studies [24,25]. Yu et al. [26] also presented a flexible tactile sensor array based on polyvinylidene fluoride (PVDF) film for measuring three-axis dynamic contact force distribution. They took advantage of similar type of sensor design and structure used in the study [23] but adopted piezoelectricity instead of capacitance as sensing mechanism. Viry et al. used conductive textile for their sensor fabrication which can detect both normal and tangential forces [27]. Alfadhel et al. introduced a magnetic nanocomposite artificial cilia implemented on magnetic microsensors for braille reading [28]. The sensor can detect both vertical and shear forces. Yuan et al. proposed a technique of vision-based optical tactile sensing of the normal, shear, and torsional load on the contact surface with a GelSight tactile sensor [29]. The system essentially measures geometry, with very high spatial resolution, which helps it to identify incipient slip as well. Recently, Wolterink et al. presented a 3D-printed piezoresistive shear and normal force sensor that can measure up to 10 N force on a fingertip phantom placed inside the sensing structure [30]. Goh et al. and Hu et al. proposed flexible and wearable pressure sensors based on the triboelectric principle [31,32]. They designed their respective sensors focusing on implementing them for a human–machine interface. He et al. also developed a piezoelectric tactile sensor array based on electrospinning and mixture process with flexible (PZT nanowires) composite films and polydimethylsiloxane (PDMS) materials claiming the potential applications of the sensor in bionics, robotics, and human–machine interaction [33]. Guo et al. demonstrated a flexible capacitive pressure sensor based on a poly(vinylidenefluoride-co-trifluoroethylene) [P(VDFTrFE)] dielectric film, a promising candidate to be employed in smart robotic skins and human–machine interfaces [34]. However, all of these sensors described can pragmatically catch forces up to two degrees of freedom, and the authors have not reported any prospects for their sensors to be utilized in an underwater environment. Brien et al. demonstrated a polyvinylidene difluoride (PVDF)-based slip sensing system to distinguish the occurrence of slippage between the fingertip and the target object and set up for use in a dexterous underwater gripper [35]. Nonetheless, most multi-directional force sensors are not capable of capturing an arbitrary force in the 3D space and only a few of them are suitable for underwater soft robotic applications.

In this paper, we propose a simple and cost-effective fabrication technique for a flexible multi-directional force sensing system. We employed four flex sensors and silicone molds to build a hemispherical shell-type sensor structure. The flex sensors are placed at 90${}^{{}^{\circ}}$ angles with one another along the circumference and aligned with the curve of the hemispherical shape. When a normal force is exerted on the top of the hemisphere, all of the flex sensors would bend homogeneously and yield nearly similar reading changes. On the other hand, when force is applied from a different direction on the sensor, flex sensors on that side of the shell would bend the most and provide unlike reading changes and distinguishable characteristic patterns. Since the flex sensors are embedded in the mold they are inherently waterproof. Thus, the fabricated sensor can be useful for an underwater soft robotic application while deciphering 3-DOF (degrees of freedom) contact forces.

## 2. Design of the Sensor Structure

Figure 1 illustrates the schematic diagram for the conceptual design of a multi-directional force sensing system with its working principle. When a normal force is exerted on the top of the sensor (90${}^{{}^{\circ}}$ latitude), the bending pattern of all the flex sensors would be the same while they would be different when the force is coming from other directions than normal and positions other than 90${}^{{}^{\circ}}$ latitude. Since the outputs from the flex sensor change based on the amount of its bending, distinctive patterns would be obtained for different directional forces applied on the sensor.

In order to develop the multi-directional force sensing system, four flex sensors are embedded in molded silicone (Dragon Skin Fx Pro (DSFP), Reynolds Advanced Materials, Boston, MA, USA) as demonstrated in Figure 1. We intend to integrate the sensor with our previously built soft robotic gripper [36]. Thus, the outer shell diameter is kept at 24 mm considering our soft robotic gripper finger thickness of 25 mm. In addition, the inner shell diameter is 12 mm so the sensor structure would be firm enough to handle a decent amount of force (5 N in our case) but not be too stiff to respond to a small range of force (0–5 N). Flex sensors are placed on an imaginary reference circle with a diameter of 20 mm so that they obtain a satisfactory amount of bending upon force while maintaining an adequate outer layer for waterproofing; however, it is to be mentioned that there is a decent amount of flexibility in choosing the materials and the dimensions of the sensor. They can be altered based on the applications of the sensor in real life.

## 3. Sensor Fabrication Process

In order to fabricate the sensor, two molds having different patterns (see Appendix A) have been generated in 3D modeling software and printed by a stereolithography (SLA) 3D printer (Form 3, Formlabs, Somerville, MA, USA). Four flex sensors (FS-L-0055-253-ST, Spectra Symbol, Salt Lake City, UT, USA) are cut into small pieces, 11.5 mm in length, so that they do not overlap with one another while molded inside the sensor structure. Overlapping of the sensors hampers obtaining distinctive output patterns and makes it complex to identify the nature of the force being applied. Clincher connectors are added to the flex sensors to solder them with the inserted thin wires shown in Figure 2a. Then, the flex sensors are clipped with the inner portion of the outer layer of pattern 1 before molding, as shown in step (v) of Figure 2b. Then, two parts of silicone (Part A and Part B) of DSFP are mixed at 1:1 proportion and poured into pattern 1 (Figure 2b(vi)). Properties of the silicone material are provided in Appendix A. A small amount of silicone tactile mutator (Slacker, Reynolds Advanced Materials, Boston, MA, USA) was added to the mold to make it softer and more flexible. The materials are chosen such that they can withstand but deform well within their elastic limit in response to a small amount of force (0–5 N). Due to the attached steel connector at the bottom of each flex sensor, there remained a thin space in between the flex sensor and the outer shell of pattern 1 even after clipping (Figure 2b(v)). This creates a thin outer layer of silicone over the flex sensor (Figure 2b(vi)), which keeps them in position aligned with the curvature of the shell when the silicone is cured and the clips are detached from the mold. Afterward, the silicone is cured for 1 hour at room temperature, and subsequently, the pattern is broken down to obtain the base structure of the sensor (Figure 2b(vii)). Next, the silicone mixture is poured almost half full into the hemispherical shaped pattern 2 and kept there for some time (5–10 min) to be cured a little. Then, the previous mold is placed in it (Figure 2c(viii)). Due to the gravity, the mold goes down but the slightly cured and highly viscous silicone keeps it floating. Thus, approximately 3–4 mm of the outer layer is formed over the flex sensors (Figure 2c(ix)). Finally, the inner mold is removed (Figure 2c(x)) and the sensor structure is punched out from the hemispherical pattern 2 (Figure 2c(xi)). Schematic illustrations for the fabrication process are demonstrated in Figure 2 and photos for the fabrication process are also shown in Appendix A. It is to be mentioned that the choice of materials and dimensions solely depends upon the applications of the force sensor. The related fabrication processes such as choice of silicone grade, silicone mixing ratio, the addition of mutator, curing time, etc., may vary from this specific case.

## 4. Results and Discussions

### 4.1. Force Calibration

Commercially available indentation equipment (AGS-X, Shimadzu, Kyoto, Japan) with a 10 N force gauge and a 10 mm diameter indenter was used to calibrate the fabricated sensor. Since the equipment is capable of indenting in the vertical direction only, the fabricated sensor was aligned and fixed in a position to apply forces on desired locations precisely. Figure 3 shows the force calibration setup for the fabricated tactile sensor.

As discussed in the conceptual working principle of the sensor, all the flex sensors would bend in a similar fashion as the normal force exerted on the top of the shell. Figure 4 demonstrates the response of the sensor for the normal force along z-axis and validates the claim. A simple voltage divider circuit was used to obtain the output reading data from the flex sensors in addition to Savitzky–Golay finite impulse response filter for a smooth data representation [37]. As the internal pressure in the hemispherical shell structure builds up due to the applied force, the flex sensors start to bend and their respective resistance values vary. Although the reading change in each of the sensors is different than others due to their dissimilar sensitivity and proper placement limitation during the fabrication, the pattern of change shows the same trend. It is important to indicate that the sensor structure is observed to encounter plastic deformation for a force of little over 6 N (Appendix A). Thus, it is safe to keep the force range between (0–5) N to make sure the deformation of the sensor is within the elastic limit of the sensor materials.

We also investigated the effect of exerting other directional forces along the x- and y-axis in both positive and negative directions and along the lines with ±45${}^{{}^{\circ}}$ increments from the x- and y-axis. The effect of applying forces along ±x- ±y-direction, and the lines with ±45${}^{{}^{\circ}}$ from the x- and y-axis are illustrated in Figure 5 and Figure 6, respectively. From the figures, the output voltage values increase when putting force on the sensor parallel to xy-plane, which is quite the opposite to what happens when a normal force is applied on the sensor along the z-axis. It can be explained by the fact that force on the sensor parallel to the xy-plane would straighten the flex sensors proximal to the force exerting location as illustrated in Figure 1c. The phenomenon is theoretically equivalent to bending the flex sensors in opposite direction (Figure 7b). Thus, the relative voltage increases as the force is being exerted on the sensor parallel to the xy-plane.

Figure 5 demonstrates that the only flex sensor which is happened to be placed in line with the force along ±x- or ±y-axis and in the vicinity to the force exertion position shows the significant reading change while force being applied on the ±x- and ±y-axis. The pattern can be utilized to check whether the force is being exerted exactly on the ±x- or ±y-axis or not. It can also be evaluated from the figures that the rates of change for certain flex sensors’ data are sharp compared to the following cases where forces are applied at an angle of ±45${}^{{}^{\circ}}$ from the x- and y-axis. The reason can be elucidated by the fact that considerably greater stress is generated in these cases due to the presence of flex sensors aligned with the x- and y-axis, and straightening of the flex sensor occurs quickly.

Figure 6 represents the impact of applying force along the lines with ±45${}^{{}^{\circ}}$ from the x- and y-axis. Forces have been applied to a position in-between two flex sensors at 30${}^{{}^{\circ}}$ latitude in parallel to the xy-plane as mentioned in the simulation section. The trend of sensor responses in this scenario is quite distinctive in nature. Among four, the readings of two flex sensors positioned at the proximity of the force diverge from the other two. The force exertion effect is almost similar to the previous cases where a force is applied along the x- and y-axis; however, the sensor response is depreciated compared to the previous cases. This is because the force is not applied directly to the flex sensors in this case and the straightening of the flex sensors takes place more gently and so does the change in the readout. This distinguishable pattern can also help assess the force location on the sensor. It is to be noted that there is a minimum limit on the size of the object that the sensor can detect precisely. If the object is too thin or sharp, the sensor might not behave properly. In that case, the object might penetrate through the sensor because of the softness of the silicone material. Another scenario would be the flex sensors not bending as discussed while the force being exerted as a normal force on the hemisphere (along the z-axis) or in between two flex sensors (along ±45${}^{{}^{\circ}}$ angle with x- or y-axis in xy-plane) because force is required to apply on a minimum area to bend the desired flex sensors for defined forces. The object size must be more than a certain limit to obtain a consistent readout from the sensor. In this study, we have considered a 10 mm diameter object for applying the forces and calibrating the sensor. The effective force detection range of the sensor may vary with the alteration of the size of the object. If the object size changes, one has to calibrate the sensor accordingly. It can be regarded as one of the limitations of the sensor.

Finally, we estimated force magnitude from the sensor data by employing a calibration matrix equation. The calibration matrix equation can be generated as follows:(1)$${\left[S\right]}_{n\times 4}\times {\left[A\right]}_{4\times 3}={\left[F\right]}_{n\times 3}$$
where,

[S] = sensor data;

[A] = compliance matrix;

[F] = force data;

*n* = no. of data points.

Sensor data can be split into four columns for four flex sensors and force matrix into three i.e., F${}_{x}$, F${}_{y}$, and F${}_{z}$; however, we applied force along the lines of ±45${}^{{}^{\circ}}$ from the x- and y-axis, which made F${}_{x}$ and F${}_{y}$ be the same value. Hence, we consider only F${}_{xy}$ in order to plot graphs instead of using F${}_{x}$ and F${}_{y}$ for those cases. During the calibration, we find the sensor outputs against the known force values. From those data, we calculate the left inverse (pseudo-inverse) of matrix [A], which is:(2)$${\left[A\right]}_{4\times 3}={\left[S\right]}_{4\times n}^{-1}\times {\left[F\right]}_{n\times 3}$$

Next, with the stored compliance matrix data [A], we evaluate the force value for given sensor data. The comparison results between the estimated force and the actual force values for three sample cases are depicted in Figure 8. In the case of loading along the x-axis, force estimation slightly under-predicts the actual case while somewhat over-predicts for the applied forces along the z-axis, and line with ±45${}^{{}^{\circ}}$ from the x- and y-axis. Here, the [A] matrix is comprised of only one set of data during calibration, which makes it susceptible to over-predict, or under-predict the force values. Although the evaluated forces show a little deviation from the actual case, they reasonably can predict the force values given the sensor data. Collecting a significant amount of data from the sensor for numerous force values and locations, and applying machine learning could possibly be another way to obtain a more robust [A] matrix and implement that to predict the force values more accurately and precisely.

### 4.2. Simulation

The deformation of the hemispherical shell structure was evaluated using the finite element method (FEM) in the commercial software, ANSYS Mechanical by Ansys Inc., Canonsburg, PA, USA. To simulate the real system, the hemispherical geometry was constructed, including the flex sensors and steel plates with soldering. The hemisphere and flex sensor dimensions were maintained exactly the same as the fabricated sensor; however, to keep the geometry relatively simple, an equivalent area on top of the flex sensor was considered for steel plate and soldering material combined. The geometry is shown in Figure 9a(i). All the material properties were assigned to the respective geometries. The material properties for the DSFP was gathered from the manufacturer’s datasheet [38]. The originally calculated Young’s modulus for DSFP was considered to be reduced by about 60% after mixing the silicone tactile mutator [39], which further softens the silicone rubber. In the case of flex sensor material properties, reasonable assumptions were made. The geometry was then meshed using unstructured mesh (Figure 9a(ii)) to perform the simulations. The ANSYS Mechanical solver utilizes the following overall equilibrium equations for linear structural static analysis [40]:(3)$$\left[K\right]\left\{u\right\}=\left\{F^{a}\right\}+\left\{F^{r}\right\}$$
where,

$\left[K\right]$ = total stiffness matrix= $\sum_{m=1}^{N}\left[K_{e}\right]$;

$\left[K_{e}\right]$ = element stiffness matrix;

*N* = number of elements;

{u} = nodal displacement vector;

$\left\{F^{r}\right\}$ = reaction load vector;

$\left\{F^{a}\right\}$ = total applied load vector.

Finally, the boundary conditions were set according to the experimental setup. The sensor base was kept fixed while forces were applied on a circular area of 10 mm diameter to replicate the force from the indenter in the experimental setup. Multi-directional forces were applied to the sensor in the experiments. The forces from two latitudes that were in the same directions described in the force calibration were considered for the simulations: (i) normal to the equator on the top pole of the hemisphere (latitude 90${}^{{}^{\circ}}$) and (ii) parallel to the equator near the base of the hemisphere (latitude 30${}^{{}^{\circ}}$). Figure 9b depicts the applied force directions for both of the cases. At latitude 30${}^{{}^{\circ}}$, forces parallel to the equator were applied on two positions: (i) in between two flex sensors and (ii) on one of the flex sensors. Only one force was considered at a time for all the simulations, creating three separate cases. The simulations were run to find out the deformation caused by the forces applied. Figure 9c shows the resulting deformations by a force of 5 N in both directions.

The force–deformation graphs for forces in the range of 0.5–5 N were generated from the simulations for all three cases, and each case was compared with the experimental results. The comparison as well as the applied force position and direction for each case are illustrated in Figure 10. Figure 10a depicts deformation due to normal force at 90${}^{{}^{\circ}}$ latitude while Figure 10b,c show deformation for two forces, one between two flex sensors and the other on one of the flex sensors, both at 30${}^{{}^{\circ}}$ latitude. From the figure, it could be concluded that the simulated model can predict the deformation of the shell with considerable accuracy. The simulations could be more accurate with more precise material properties for each of the materials. The small deviations from the experimental results could also be explained by imperfect fabrication resulting in slight changes in the geometry and symmetry. It could also be observed from Figure 10 that the forces applied in the experiments were within the elastic limit of the soft sensor.

### 4.3. Repeatability Test

In addition to the fundamental physical properties of the tactile force sensor, precision is one of the most important factors in determining sensor performance [41]. In order to check the repeatability of the sensor readings, we acquired the dataset from the three different cases of applied force ((a) along the z-axis, (b) along the x- and y-axis, and (c) along the lines with 45${}^{{}^{\circ}}$ from the x-axis) with three separate trials per case. The same indentation technique was adopted for multi-direction force to check the repeatability test. Appendix A and Figure 11 demonstrate the repeatability test results for individual trial and combined cases accordingly. Since the sensor structure is hemispherical, the sensor response for other force directions parallel to the xy-plane can be regarded as interchangeable. It can be seen from the figure that the trend of the sensor responses for a specific directional loading are quite consistent excluding minute deviations. In response to the normal force along the z-axis, sensor outputs always go down as the force increases. Particular sensor output (here, F4) on which the force being exerted shows the substantial reading change for force along the y-axis case. In addition, two of the sensors (here, F1 and F4) manifest a similar pattern of rising when forces are acting in between them. Ideally, F1 and F4 should show the identical readings while a force is being applied exactly at the 45${}^{{}^{\circ}}$ angle in-between them; however, there are discrepancies from the theory due to the limitation of the fabrication method of placing each of the flex sensors perfectly in the alignment with one another and the different sensitivity of the individual flex sensor. Nonetheless, their patterns of change are identical irrespective of samples and trials. It corroborates the quality of the coherence of the sensor.

### 4.4. Cyclic Loading and Hysteresis Test

To determine whether there would be any hysteresis in the sensor after some loading cycle, we conducted another experiment in which a cyclic normal force is applied along the z-axis on the sensor up to 5 N and then kept at that state roughly for 30 s. Subsequently, force is released and maintained at zero for a while before the next period begins. The cycle is continuously replicated 10 times manually. The sensor output over the ten cycles for the cyclic loading on the sensor is demonstrated in Figure 12a. The corresponding force–displacement graph for the hysteresis test is shown in Figure 12b, where the average and the standard deviation of force values over the ten cycles of the cyclic loading for a certain displacement are represented. Additionally, force–displacement curves for all the ten cycles are presented in Appendix A. The maximum discrepancy measured during the loading–unloading cycle is found to be 0.52 N in average at 2.5 mm displacement which is 10.4% of the force range, 5 N. The area of the the hysteresis loop is 1.2 × 10${}^{-3}$ J in average (16.5% of the elastic energy stored during the loading cycle) which refers to the amount of energy dissipated during a cycle. It is to be mentioned that the loading data are machine tabulated and unloading ones are taken manually, which causes a greater standard deviation during the unloading cycle of the test (Figure 12b). As discussed in the previous sections, the sensor is working pragmatically within the elastic limit of the material DSFP. It can be elucidated from Figure 12 that the sensor is fairly robust against and does not encounter significant hysteresis since the area within the hysteresis loop is small and the initial values of the sensors do not diverge and can be regenerated even after a certain period of cyclic loading. These imply the prospects of the fabricated sensor for further investigation and the possibility of employing it in industrial use as well as research purposes.

### 4.5. Localization of Point of Contact

We consider single-point force measurement in order to configure the applied force location on the sensor. For simplicity in the calculation, we take the projection of the hemisphere on the xy-plane and divide it into the 3×3 grid. We place an index on each grid according to its row and column number (see Appendix A). Based on the trend of the flex sensors data, we specify the grid to be activated in response to the force. First, we form a row matrix for all the flex sensors and set the value of each item to zero. Next, we define a threshold value for all the flex sensors to pass. Readings from the flex sensors for applied force are fed, and if certain sensor reading passes the threshold, the flex sensor value would convert from zero to one in the row matrix. In that way, we would obtain a definite row matrix for each of the cases. In addition, utilizing the matrix, a decision for activating a specific grid can be taken. A table (see Appendix A) has been provided in the Appendix A regarding the row matrix and grid activation. Figure 13 illustrates the response of the sensor for different forces and corresponding applied force locations on the sensor. It is to be noted that there has to be a certain amount of load on the sensor to pass the predefined threshold value and to specify the location of the force; however, we aim to implement a machine learning algorithm to better assess the trend of the readings from the flex sensors for different types of loading in various locations to increase the identification responsiveness of the force location as well as to better quantify their directions in three dimensions in the future scope of our study.

### 4.6. Underwater Test

The sensor system is inherently waterproofed since the flex sensors used here are embedded in the molded silicone. In order to verify the claim, we consider testing the sensor underwater in the lab. We kept the sensor underwater (depth less than 0.5 m) for around 30 min and tested its signal. The sensor can provide signals against exerted force while remaining underwater. Figure 14 exhibits the process we carried out our test and the time-dependent sensor response for the applied force along the ±x- and ±y-direction in the sensor system coordinate. A real-time video in the Appendix A is available to demonstrate the underwater test. The successful underwater test indicates its potential applicability for underwater exploitation. According to the International Standard (ISO 20653), the IP67 standard means that the tested object is dust-tight and can go through temporary submersion without the water causing any harmful effects [42,43,44]. Our sensor has been tested underwater, and it could reasonably achieve the IP67 level protection for now. Since we intend to utilize the sensor with our lab-made soft robotic gripper [36] for underwater manipulation, we are actively working on the modification of the fabrication methodology and enhancing the performance of the sensor for underwater applications. We are also trying to make the sensor more robust against water so that it can go through continuous submersion to achieve the IP68 standard.

## 5. Conclusions

In recapitulation, we have designed and fabricated a flex sensor-based multi-directional force sensing system employing a simple and low-cost methodology. The force sensing system can effectively detect and estimate forces from multiple directions. A single point of contact on the sensor has been identified on a two-dimensional 3×3 grid. The sensor outputs were repeatable for multi-directional forces without noticeable hysteresis detected during the cyclic loading test. Moreover, the sensor showed its waterproofness as the feasibility of potential usages in underwater applications. So far, we have reasonably estimated the protection level of the sensor to be comparable to the IP67 standard, and are working on its enhancement. These favorable results signify the promising outlook of the sensor to be implemented in soft robotics, manufacturing industries, research purposes specially in biological sampling in an underwater environment, and so on.

The future direction of the study would be to map the sensor against the force by employing a machine learning algorithm so that it can generate a force vector (including the direction, magnitude, and location) in real-time in response to a general force applied on the sensor. We also plan to integrate the sensor with our previously built soft robotic gripper [36] to check its applicability for tactile information gathering, especially in an underwater environment. Moreover, employing other types of smart materials such as electroactive polymers (EAP) or piezoelectric materials to fabricate a multi-directional soft force sensor with competitive sensitivity would be another possible direction of the study.

## Figures and Tables

**Figure 1 sensors-22-03850-f001:**
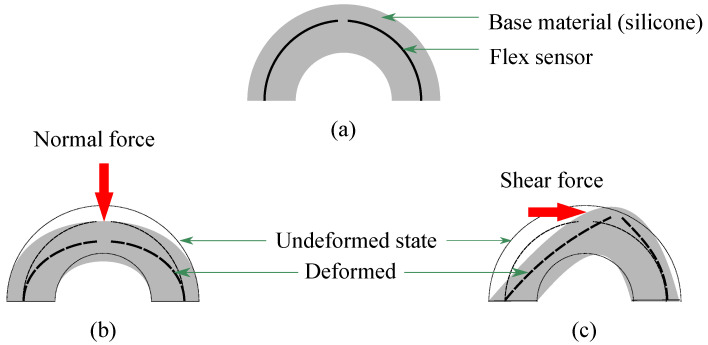
Schematic diagram of the (**a**) sensor structure, and working principle of the multi-directional force sensing system against (**b**) normal and (**c**) shear force; red arrows depict applied force directions.

**Figure 2 sensors-22-03850-f002:**
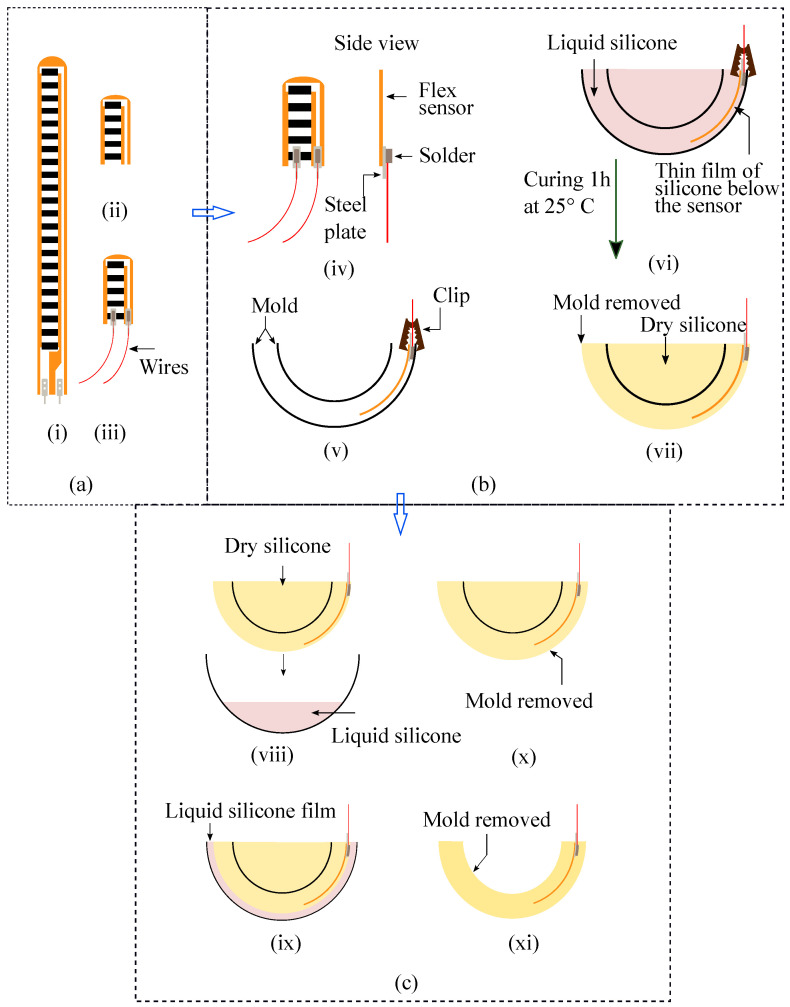
Schematic diagram of the step-by-step fabrication process of the flex sensor-based multi-directional force sensing system: (**a**) flex sensor preparation; (**b**) placing the flex sensors inside the mold; (**c**) creating a thicker outer layer over the flex sensors.

**Figure 3 sensors-22-03850-f003:**
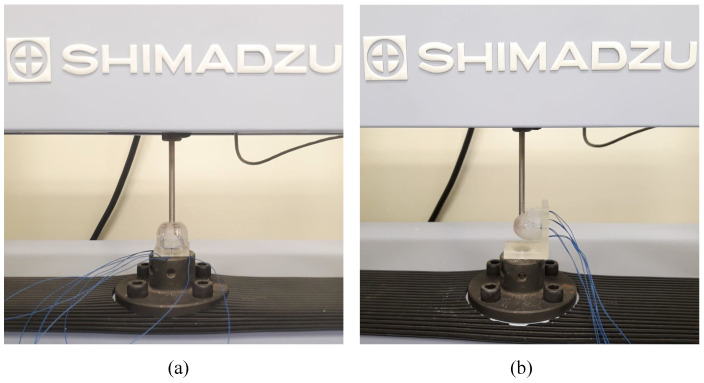
Force calibration setup for the multi-directional tactile sensor: (**a**) force along z-axis and (**b**) force exerted from the different directions of xy-plane.

**Figure 4 sensors-22-03850-f004:**
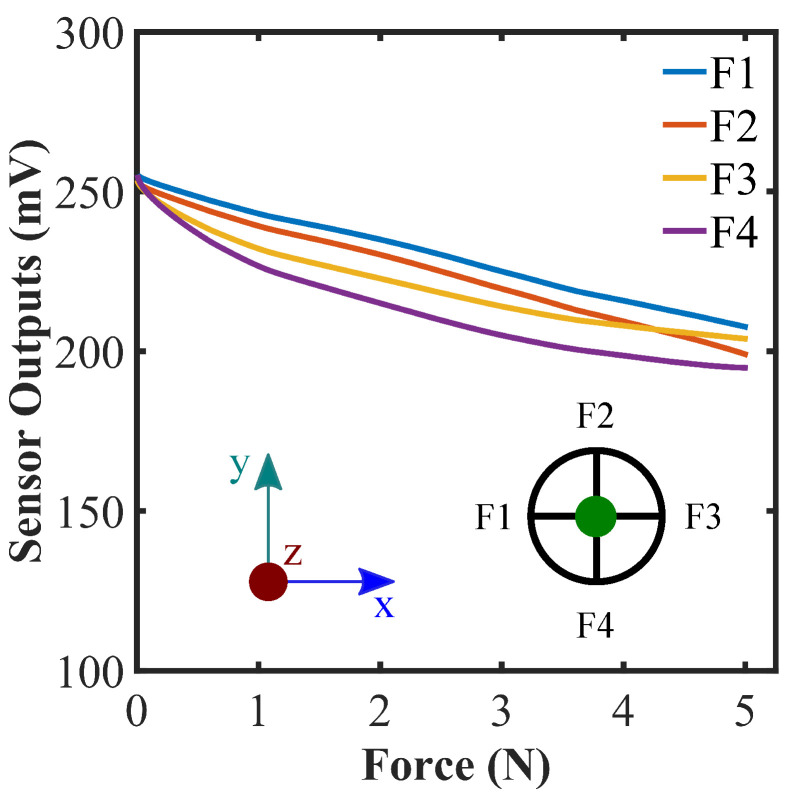
Force calibration against the normal force on the top of the hemispherical sensor; top view of the sensor system is shown, where four flex sensors are numbered as F1, F2, F3, and F4 accordingly. The green dot represents the force direction along z-axis into the page.

**Figure 5 sensors-22-03850-f005:**
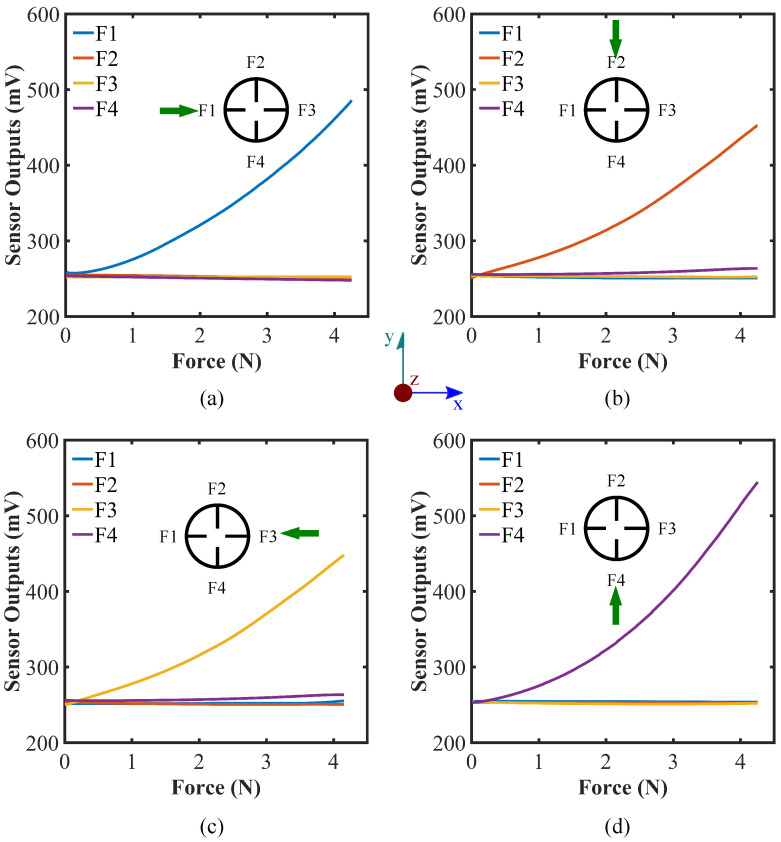
Force calibration against forces on the sides (specifically on the inserted flex sensors) of the hemispherical sensor (at 30${}^{{}^{\circ}}$ latitude), along (**a**) x-axis, (**b**) -y-axis, (**c**) -x-axis, and (**d**) y-axis; top view of the sensor system is illustrated and the green arrow depicts the applied force directions.

**Figure 6 sensors-22-03850-f006:**
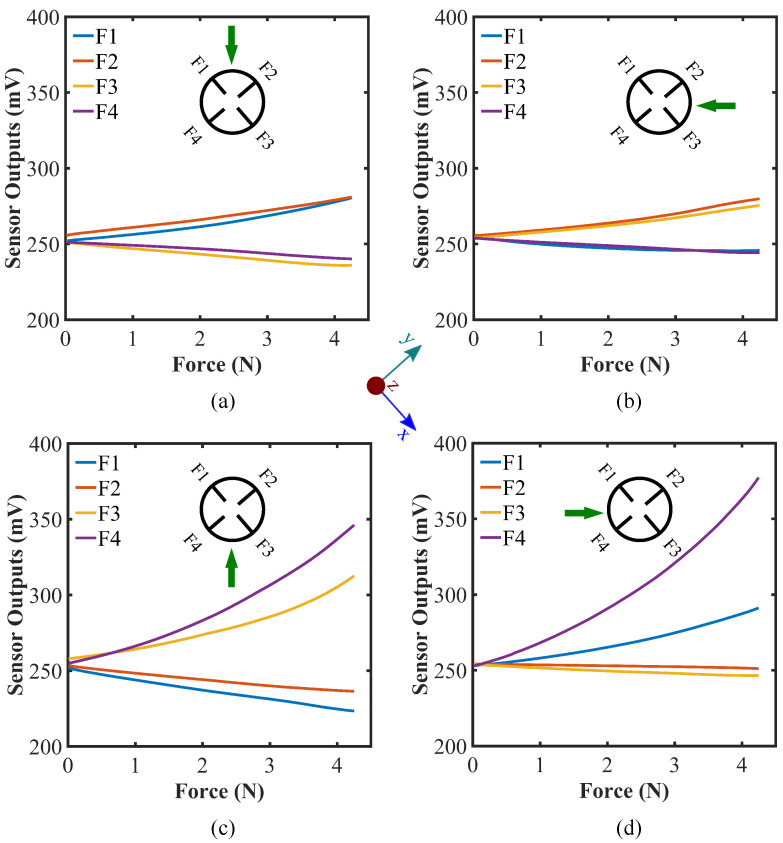
Force calibration against forces on the sides of the hemispherical sensor (at 30${}^{{}^{\circ}}$ latitude),(in-between the inserted flex sensors) with 45${}^{{}^{\circ}}$ along (**a**) -y-axis, (**b**) -x-axis, (**c**) y-axis, and (**d**) x-axis (measuring angle clockwise); top view of the sensor system is shown and the green arrow indicates the exerted force directions.

**Figure 7 sensors-22-03850-f007:**
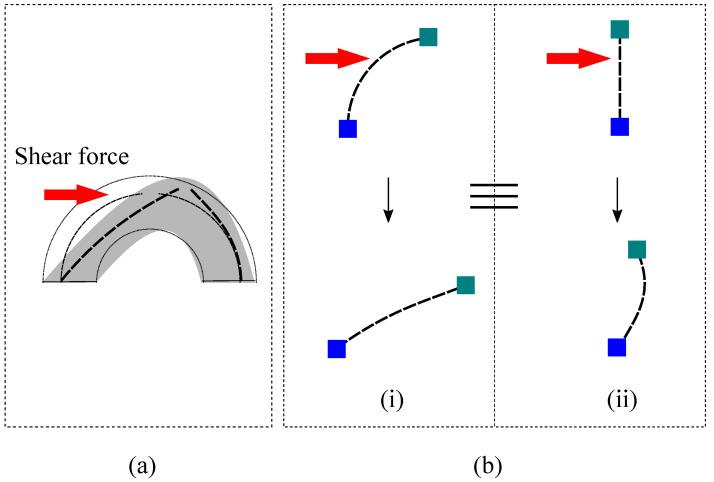
Effect of applying (**a**) shear force on the sensor (**b**) equivalent systems; blue squares represent fixed and greenish ones portray partially moving boundary conditions; in both of the cases, red arrows symbolize applied force direction.

**Figure 8 sensors-22-03850-f008:**
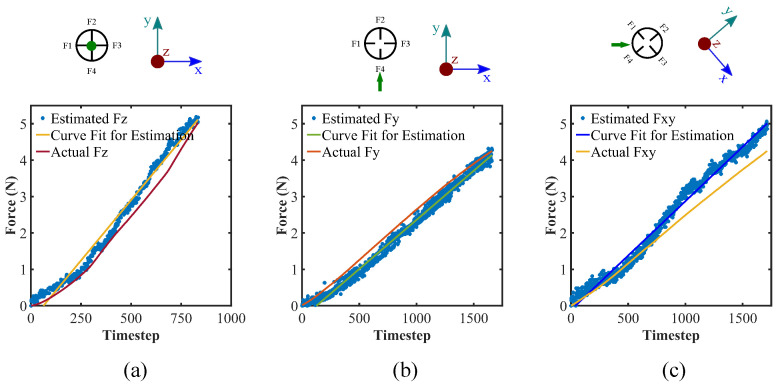
Force estimation utilizing sensor data and calibration matrix; applied force direction along (**a**) z-axis, (**b**) y-axis, and (**c**) line with 45${}^{{}^{\circ}}$ from the x-axis.

**Figure 9 sensors-22-03850-f009:**
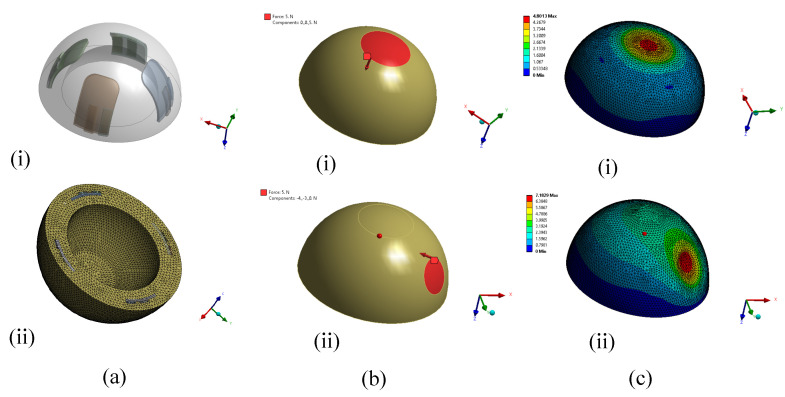
Simulation steps to evaluate the sensor shell deformation: (**a**) (i) geometry of the sensor and (ii) mesh; (**b**) direction of the applied forces: force parallel to (i) z-axis and (ii) xy-plane; (**c**) total deformation contour as the result of applied force along (i) z-axis and parallel to (ii) xy-plane.

**Figure 10 sensors-22-03850-f010:**
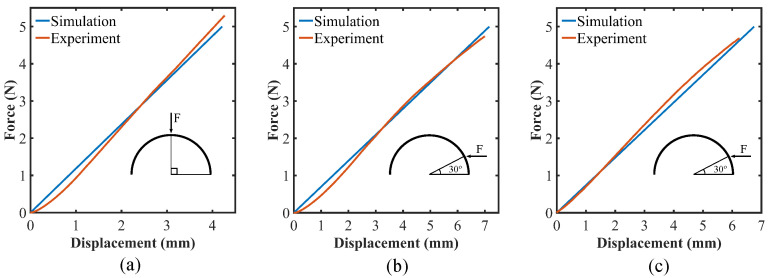
Sensor deformation for applied force (**a**) at 90${}^{{}^{\circ}}$ latitude; (**b**) at 30${}^{{}^{\circ}}$ latitude, between two sensors; (**c**) at 30${}^{{}^{\circ}}$ latitude, on one of the sensors.

**Figure 11 sensors-22-03850-f011:**
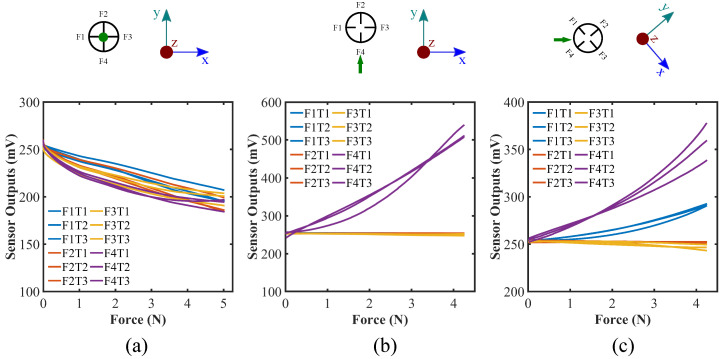
Repeatability test for force calibration for the forces along (**a**) z-axis, (**b**) y-axis, and (**c**) 45${}^{{}^{\circ}}$ angle from the x-axis; sensor responses for all three trials for each case are shown in the graphs accordingly; green dot and arrow represent the force directions; top view of the sensor system is also presented.

**Figure 12 sensors-22-03850-f012:**
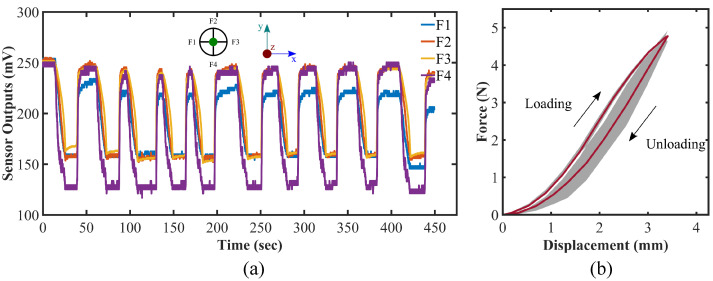
(**a**) Cyclic loading test and (**b**) hysteresis test results; green dot in (**a**) representing the exerted force direction along z-axis, and shaded area in (**b**) depicting the standard deviation of force for a certain displacement value.

**Figure 13 sensors-22-03850-f013:**
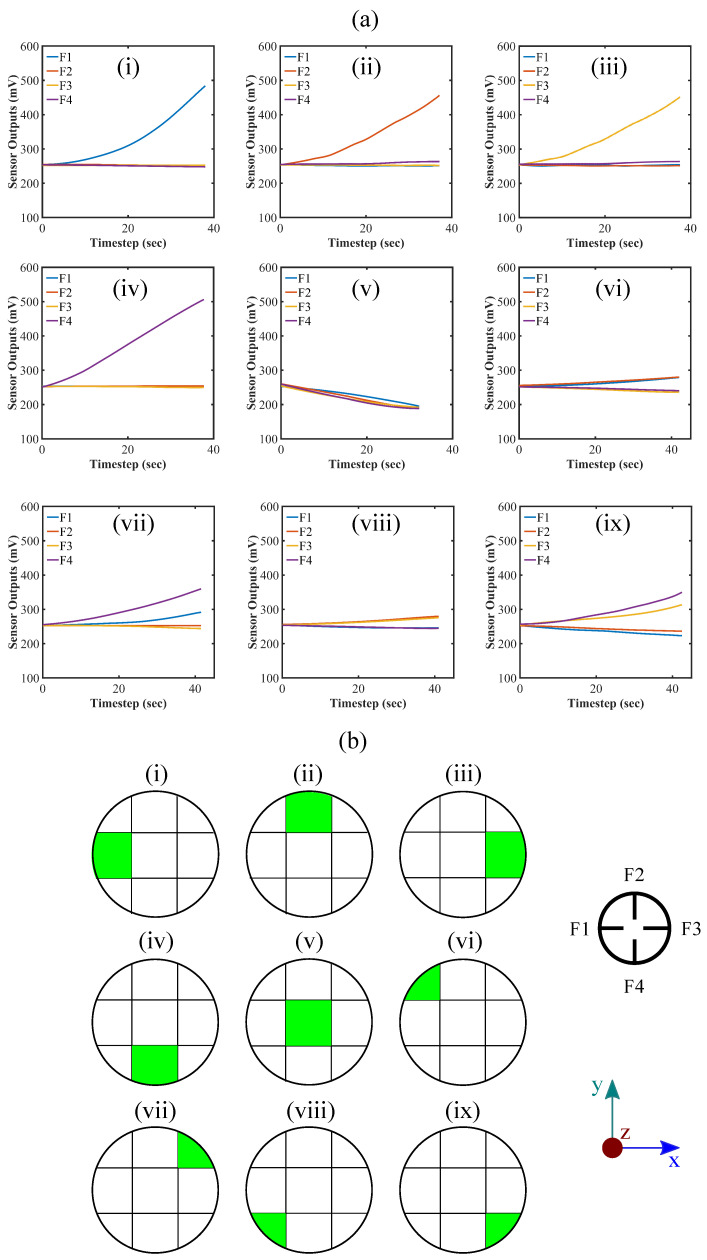
Exerted force location recognition in the grid: (**a**) sensor response due to the applied force and (**b**) corresponding force location generation; top view of the sensor system and coordinate axis are depicted in the lower right part.

**Figure 14 sensors-22-03850-f014:**
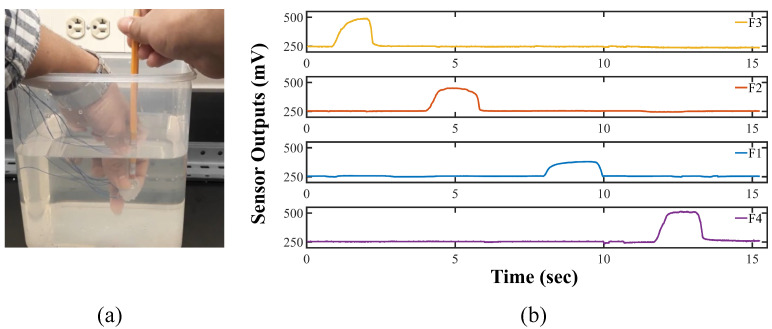
Underwater test for the sensor (**a**) exerting forces and (**b**) corresponding readings of the sensor.

## Data Availability

All data are available upon request.

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
