# Peer review of "Soft Multi-Directional Force Sensor for Underwater Robotic Application"

_sensors, 2022, doi:10.3390/s22103850_

Round 1

Reviewer 1 Report

They test the prototype in real conditions in real underwater robotic applications.

Reviewer 2 Report

In this paper, a flexible multi-directional force sensing system is proposed, which is also favorable to be utilized in underwater environments. The force sensing system can effectively detect and estimate forces from multiple directions. The favorable results signify the promising outlook of the sensor to be implemented in soft robotics, manufacturing industries, research purposes specially in biological sampling in an underwater environment, and so on. The concept proposed in this paper is very interesting, and this paper is well written and organized. I think this work may be accepted after addressing the following revisions:

  1. This work explained details in the tactile stimuli transition with the shape deformation, and the voltage distribution at different position of force sensing system when touching objects. Will the size of the objects affect the sensing performance?

  1. Whether force sensing system has been tested in the sea, seawater will or not corrode the sensor and affect the accuracy of the sensor. Please add the corrosion resistance test of force sensing system. What is the advantage of force sensing system over a simple optical camera with a light source?

  1. The response is similar against different changing factors. Authors have an idea to distinguish the responses against different changing factor.

  1. Figure 7 is not clear, authors may present a cleared window of this figure as the supplement.

  1. Will the mechanical properties of the flexible cover affect the performance of force sensing system? The author can give the readers some suggestions on material selection.

  1. How to solve the problem of sensor packaging? What is the detection range and limit of flexible multi-directional force sensing system?

  1. There are some similar multi-directional force sensing system before, like Smart Mater. Struct. 30, 075012 (2021)and 9864967 (2021). Authors should provide comparisons and emphasize their novelty in the introduction.

Reviewer 3 Report

3D physical sensors using EAP materials have been developed in various forms, and technically it is difficult to find special differences.

Chapter 3 describes the manufacturing process of the sensor in great detail, but it is difficult to generalize and is only a reference for the experimental manufacturing process of the sensor.
That is, detailed values ​​in the process, such as the curing time of the sensor material, may vary depending on the physical properties of the material, such as the curing agent mixing ratio, electrical conductivity, and the type of polymer.

In particular, the process technology presented in this paper is at a level that can be simplified and miniaturized if there is sufficient experience and knowledge about EAP materials.

Therefore, it is judged that it is more helpful to describe material properties and electrical and physical properties rather than presenting detailed figures and processes.

In particular, direction and repeatability were tested in the sensor test, and the linearity of each section was also confirmed. However, it appears that the test for the hysteresis effect is missing in the important characteristics of the sensor, and it is necessary to supplement it.

However, I think it is meaningful to present a sensor technology that can be used in water with a simple structure.

Round 2

Reviewer 2 Report

The authors have carefully revised the manuscript according to the reviewer's comments. In my opinion, the revision can be accepted.

Reviewer 3 Report

The author argues that as a typical underwater sensor uses a camera. However, most underwater robots use cameras in the visible area for recording and environment recognition due to turbidity or the specificity of the underwater environment.
Also, tactile sensors know that their detection area is very local, so ultrasound scanners and other physical sensors are more often used. In other words, it does not seem appropriate to express a non-universal fact as a general fact.

And, the developed sensor claims to be a sensor with high sensitivity in three axes that can be used underwater.
However, verification of the hysteresis effect, which is one of the main characteristics of the sensor that was questioned in the previous review, is insufficient. If the hysteresis is large, it is disadvantageous to use it as a sensor, so it is necessary to identify the clear physical characteristics of the sensor in advance and suggest the possibility of use at a level that can be corrected.
Another important issue is that the effect of water pressure cannot be neglected when used underwater. When water pressure is applied in the water according to the depth of the water, the external pressure will increase uniformly in all directions, but the section that guarantees the linearity of the sensor may change, so environmental conditions must be specified.
In addition, it appears that it is necessary to analyze the physical and electrical characteristics more accurately than the manufacturing process of the sensor pointed out last time.

Round 3

Reviewer 3 Report

As a result of the review, a description of the sensor's limitations and hysteresis effect pointed out last time was added, and directions for use in the water were also explained. It was also explained that a general approach considering the properties of EAP materials is possible.
Although the explanation and supplementation of the points pointed out was well done, as an academic research thesis, it would be better to explain more quantitative experimental results or simulation results numerically. In particular, I think it would be better to express the hysteresis as a numerical value rather than simply explaining that it has a small effect because it has a small area.
It is judged appropriate to suggest the author's efforts to improve the completeness of the thesis and the direction for sensor production and utilization.
